# The ULK1/2 and AMPK Inhibitor SBI-0206965 Blocks AICAR and Insulin-Stimulated Glucose Transport

**DOI:** 10.3390/ijms21072344

**Published:** 2020-03-28

**Authors:** Jonas R. Knudsen, Agnete B. Madsen, Kaspar W. Persson, Carlos Henríquez-Olguín, Zhencheng Li, Thomas E. Jensen

**Affiliations:** 1Section of Molecular Physiology, Department of Nutrition, Exercise and Sports, University of Copenhagen, 2100 Copenhagen, Denmark; jrk@nexs.ku.dk (J.R.K.); agbmadsen@gmail.com (A.B.M.); plj868@alumni.ku.dk (K.W.P.); col@nexs.ku.dk (C.H.-O.); zxk997@alumni.ku.dk (Z.L.); 2Laboratory of Microsystems 2, Institute of Microengineering, Ecole Polytechnique Fédérale de Lausanne, 1015 Lausanne, Switzerland

**Keywords:** AMPK, ULK1/2, skeletal muscle, kinase inhibitor, SBI-0206965

## Abstract

The small molecule kinase inhibitor SBI-0206965 was originally described as a specific inhibitor of ULK1/2. More recently, it was reported to effectively inhibit AMPK and several studies now report its use as an AMPK inhibitor. Currently, we investigated the specificity of SBI-0206965 in incubated mouse skeletal muscle, measuring the effect on analog 5-aminoimidazole-4-carboxamide ribonucleotide (AICAR)-stimulated AMPK-dependent glucose transport and insulin-stimulated AMPK-independent glucose uptake. Pre-treatment with 10 µM SBI-0206965 for 50 min potently suppressed AICAR-stimulated glucose transport in both the extensor digitorum longus (EDL) and soleus muscle. This was despite only a modest lowering of AICAR-stimulated AMPK activation measured as ACC2 Ser212, while ULK1/2 Ser555 phosphorylation was prevented. Insulin-stimulated glucose transport was also potently inhibited by SBI-0206965 in soleus. No major changes were observed on insulin-stimulated cell signaling. No general effect of SBI-0206965 on intracellular membrane morphology was observed by transmission electron microscopy. As insulin is known to neither activate AMPK nor require AMPK to stimulate glucose transport, and insulin inhibits ULK1/2 activity, these data strongly suggest that SBI-0206965 has a non-specific off-target inhibitory effect on muscle glucose transport. Thus, SBI-0206965 is not a specific inhibitor of the AMPK/ULK-signaling axis in skeletal muscle, and data generated with this inhibitor must be interpreted with caution.

## 1. Introduction

The Unc-51 like autophagy activating kinases (ULK)1/2 are best known as being essential regulators of the initiation of autophagy [1]. ULK1/2 are themselves regulated by phosphorylation by upstream kinases AMP-activated protein kinase (AMPK) and mechanistic target of rapamycin complex 1 (mTORC1) [2]. The catabolism-promoting AMPK-complex is canonically activated in response to energy deficiency by an increased AMP/ATP ratio and perhaps also AMP-independently by glucose starvation [3], and phosphorylates ULK1/2 on Ser555 to increase its activity [4]. In skeletal muscle, AMPK-dependent phosphorylation of ULK1/2 has been suggested to regulate autophagy in the contexts of exercise and starvation [5,6]. In skeletal muscle, AMPK is also a mediator of glucose transport induced by the AMP analog 5-aminoimidazole-4-carboxamide ribonucleotide (AICAR) or alternative AMPK activating compounds [7,8].

Currently, we set out to use the small-molecule inhibitor SBI-0206965 in our experimental setup, incubated muscle, to tease out the role of the AMPK/ULK1/2 signaling axis in the regulation of glucose transport and autophagy. SBI-0206965 was reported as the first ever small-molecule inhibitor of ULK1/2 in a study by Egan et al. in 2015 [9]. This compound was reported to have an IC50 of 108 and 711 nM against ULK1 and 2, respectively, and only inhibiting 10 out of 456 tested kinases >95% when tested in a kinase panel in vitro at 10 μM. In 2018, Dite et al. reported that SBI-0206965 was a very potent inhibitor of AMPK, reporting IC50 values of 1.05, 0.40, and 0.33 µM against ULK1, α1 AMPK, and α2 AMPK, respectively, concluding SBI-0206965 to have “utility as a tool compound for investigating physiological roles for AMPK” [10]. Consistent with Egan et al., SBI-0206965 seemed fairly selective, inhibiting only 5 of 50 kinases tested by >50% at 0.25 µM. Some articles have since been published using SBI-0206965 to investigate the role of AMPK in the regulation of various biological endpoints [11,12].

During our initial testing, however, we found SBI-0206965 to be an extremely potent inhibitor of muscle glucose transport stimulated by AICAR, despite only modest phosphorylation impairments in the AICAR-induced signaling. Furthermore, we observed a similar effect of SBI-0206965 on insulin-stimulated glucose transport. As insulin regulates glucose transport independently of AMPK and inhibits ULK1/2, this is likely an off-target effect, highlighting the importance of the critical use of this compound to study AMPK/ULK signaling-regulated endpoints.

## 2. Results

In murine embryonic fibroblast cells, 10 µM SBI-0206965 for 1 h was sufficient to inhibit ULK1/2 kinase activity [9]. Thus, this dose and time was chosen for our initial testing in incubated adult mouse skeletal muscle. Using the same concentration and incubation time, we observed a variable inhibition of unstimulated glucose transport, and a potent inhibition of AICAR-stimulated glucose transport into glycolytic extensor digitorum longus (EDL) and oxidative soleus muscles (Figure 1A,B).

Despite this marked effect of SBI-0206965 on glucose transport, the effect on basal and AICAR-stimulated AMPK Thr172 phosphorylation did not reach statistical significance (Figure 2A,B), whereas Acetyl Coenzyme A carboxylase (ACC) Ser212 phosphorylation was partially reduced by SBI-0206965 in both EDL and soleus muscle (Figure 2C,D). AICAR only increased ULK Ser555 phosphorylation in DMSO-treated EDL and soleus muscles, indicating a more complete blockade of the signaling by SBI-0206965 at this level (Figure 2E,F). Representative Western blots are shown in Figure 2G. Taken together, this shows that SBI-0206965 almost prevents AICAR-stimulated muscle glucose transport at a dose where only a modest inhibitory effect on AMPK signaling is discernible.

Insulin does not activate AMPK but potently inhibits ULK activity and stimulates glucose transport into adult mouse muscle independently of AMPK [13]. However, insulin-stimulated glucose transport in incubated mouse soleus muscle was strongly suppressed by SBI-0206965 (Figure 3A). This effect was observed without any effect on the insulin-induced Akt Thr308 phosphorylation (Figure 3B) and a slight but significant reduction in Akt Ser473 phosphorylation (Figure 3C). Further downstream, no changes were observed in the insulin-induced phosphorylation of TBC (Tre-2, BUB2, CDC16) domain-containing protein family member 4 (TBC1D4) Thr642 (Figure 3D). The insulin-stimulated phosphorylation of the ULK Ser757 site was unchanged by SBI-0206965 in EDL, while the basal ULK Ser757 phosphorylation was slightly elevated by SBI-0206965 (Figure 3E), indicative perhaps of an unspecific effect on mTORC1 signaling. Representative Western blots are shown in Figure 3F. The effect of SBI-0206965 on insulin-stimulated glucose transport strongly suggests that this is an unspecific off-target effect unrelated to AMPK and ULK1/2 inhibition.

ULK1/2 is known to signal via the Vacuolar protein sorting (VPS)34) complexes involved in autophagy and endocytic sorting [14,15] to initiate autophagy [16]. Mice with muscle-specific KO of the obligate VPS34 partner, VPS15, display massive accumulation of vacuoles with varying membrane layers and autophagosomes, as well as increased mitophagy [17]. We suspected that SBI-0206965-induced ULK1/2 inhibition might cause similar gross changes in intracellular membrane accumulation and morphology, thereby disrupting overall vesicle formation and trafficking, including that of glucose transporter 4 (GLUT4). Such an effect would be predicted to disrupt both AICAR and insulin-stimulated muscle glucose transport. We, therefore, evaluated soleus and EDL muscles incubated with or without SBI-0206965 by transmission electron microscopy (TEM). We were able to identify the lamellated vacuoles previously described to accumulate in VPS15 KO muscles [17], both in muscles treated with or without SBI-0206965 (Figure 4A) These vacuoles were observed in the intramyofibrillar and perinuclear regions in both soleus and EDL muscles (Figure 4B–E). However, we did not observe any substantial changes in intracellular membrane morphology or accumulation in the presence of the inhibitor. The TEM analysis revealed active autophagy (defined as two membrane layers enclosing degraded material or organelles), including mitophagy (Figure 4F) as previously reported [17]. Autophagosomes were observed in both soleus and EDL muscles irrespective of treatment, and no major perturbation of autophagosome occurrence was detected in the presence of SBI-0206965. Overall, we did not find evidence to suggest that SBI-0206965 induces accumulation of swollen vacuoles or excessive autophagy. Thus, these data do not provide any indications that such gross impairments in the membrane trafficking system could be an explanation for the reduction in glucose transport in soleus and EDL muscles treated with the SBI-0206965 inhibitor.

To elucidate whether SBI-0206865 inhibited GLUT4 translocation, we incubated L6 muscle cells stably overexpressing GLUT4 with an exofacial c-myc tag [18] with increasing doses of SBI-0206965 with or without insulin. Interestingly, no effect of SBI-0206965 on cell surface-exposed GLUT4 was observed (Figure 5). This suggests that the effect of SBI-0206965 on glucose transport is independent of GLUT4 translocation.

## 3. Discussion

The current study shows in incubated mouse skeletal muscle that the AMPK/ULK inhibitor SBI-0206965 potently inhibits muscle glucose transport stimulated by both the AMPK activator AICAR and insulin. From our data, it is difficult to determine if the inhibitory effect of SBI-0206965 on AICAR-induced glucose transport is causally linked to the partially lowered AMPK signaling or is mechanistically independent of AMPK and ULK1/2 signaling. Nevertheless, due to the large body of evidence showing that insulin increases muscle glucose transport AMPK-independently and inhibits ULK1/2 activity, and our data showing that SBI-0206965 does not impair insulin-stimulated GLUT4 translocation, we strongly suspect that the effect of SBI-0206965 on both AICAR and insulin-stimulated glucose transport is independent of AMPK and ULK1/2. General intracellular membrane morphology was not affected by SBI-0206965, suggesting that SBI-0206965 does not mediate its effects via a generalized disturbance of intracellular vesicle trafficking, consistent with the lack of effect on GLUT4 translocation.

SBI-0206965 is not the first kinase-inhibitor shown to non-specifically inhibit muscle glucose transport independently of GLUT4 translocation. A good documented example of this is the p38 mitogen-activated protein kinase (MAPK) inhibitor SB-203580. Initially, data produced with this inhibitor were suggested to indicate that insulin regulates GLUT4 intrinsic activity. More specifically, it was shown that SB-203580 lowered insulin and contraction-stimulated glucose transport without blocking GLUT4 translocation to the plasma membrane in 3T3L1 adipocytes, L6 myotubes, and incubated rat EDL muscles [19,20]. However, a follow-up study in L6 myotubes overexpressing a SB-203580-resistant mutant of p38 MAPKα showed that SB-203580 likely directly blocked GLUT4 transport activity independent of p38 MAPK inhibition [21]. We suspect, based on our results, that SBI-0206965 might have a similar direct effect on GLUT4 transport activity. Similar to what was concluded for p38 MAPK [21], it remains a formal, albeit unlikely, possibility that the unstimulated activity of ULK1/2, rather than a stimulus-dependent increase in their activity, is required for muscle glucose transport regulation.

In conclusion, the AMPK and ULK1/2 inhibitor SBI-0206965 strongly impairs muscle glucose transport stimulated by AICAR and insulin. This is likely an unspecific effect as insulin-stimulated glucose transport is AMPK-independent and insulin inhibits ULK1/2 activity. Therefore, SBI-0206965 should not be used as a stand-alone tool to inhibit AMPK/ULK-dependent endpoints in skeletal muscle and likely in other cell and tissue types.

## 4. Materials and Methods

### 4.1. Animals

Wildtype female C57BL/6NTac mice (Taconic Europe, Lille Skensved) used in the experiments were 10–12 weeks old and littermates. The mice were housed according to the Danish legislation on animal experiments. All experiments were approved by the Danish Animal Experiments Inspectorate (2018-15-02-02-00151 granted to Erik A. Richter) and conducted in agreement with the declaration of Helsinki.

### 4.2. Muscle Incubations

Soleus and extensor digitorum longus (EDL) muscles were excised from anesthetized mice (6 mg pentobarbital and lidocaine/100 g body weight, intraperitoneal injection) and suspended in incubation chambers (Multi Myograph System, Danish Myo-Technology, Hinnerup, Denmark) containing 30 °C Krebs Ringer Henseleit (KRH) buffer supplemented with 8 mM mannitol and 2 mM pyruvate as previously described [22]. The muscles were pre-incubated for 20–40 min with the ULK1/2 inhibitor, SBI-0206965 (#18477, Cayman Chemical Company, Ann Arbor, MI, USA) or DMSO (Sigma) as a control, and subsequently stimulated with either 60 nM insulin (Actrapid, Novo Nordisk) for 20 min or 4 mM 5-aminoimidazole-4-carboxamide ribonucleotide (AICAR (Toronto Research Chemicals, North York, ON, Canada)) for 40 min with or without the inhibitors present. The total duration of the incubation experiments was 1 h.

### 4.3. 2-Deoxyglucose (2DG) Transport

2DG transport was measured for the last 10 min of the incubation experiments by using ^3^H and ^14^C radioactive tracers, as described previously [23]. In brief, the medium was changed to fresh medium containing radioactively labelled 2-[^3^H] deoxyglucose (2DG; 0.13 μCi/mL in 1 mM non-radiolabeled 2DG) and [^14^C] mannitol (0.11 μCi/mL in 8 mM non-radiolabeled mannitol) but with the treatments kept the same. The muscles were harvested by quickly washing them in ice-cold KRH buffer and snap freezing them in liquid nitrogen. While kept frozen, they were trimmed from visual tendons, weighed, and lysed as described below. Then, 100 μL of the lysate was dissolved in 2 mL of β-scintillation liquid (Ultima Gold, Perkin Elmer). In addition, 2-[^3^H] deoxyglucose accumulation was used to estimate glucose transport after correction for muscle weight and extracellular space using [^14^C] mannitol.

### 4.4. Muscle Lysis

The tissue was homogenized in ice-cold Lysis B buffer (0.05 M Tris Base pH 7.4, 0.15 M NaCl, 1 mM EDTA and EGTA, 0.05 M NaF, 5 mM sodium pyrophosphate, 2 mM Na_3_VO_4_, 1 mM dithiothreitol and benzamidine, 0.5% protease inhibitor cocktail (P8340, Sigma Aldrich) and 1% NP-40), or MG buffer (10% glycerol, 1% NP-40, 50 mM HEPES pH 7.5, 150 mM NaCl, 20 mM β-glycerophosphate, 1 mM EDTA and EGTA, 10 mM NaF, 20 mM sodium pyrophosphate, 2 mM Na_3_VO_4_, 10 µg/mL leupeptin and aprotinin, 2 mM phenylmethylsulfonyl fluoride, 3 mM benzamidine) for 1 min at 30 Hz, using steel beads and a Tissue Lyser II (Qiagen). Supernatants were collected after rotating the homogenates end-over-end for 45 min at 4 °C and centrifugation (18,327 g) for 20 min at 4 °C

### 4.5. Western Blotting

Total protein concentration of the lysates was determined by bovine serum albumin (BSA) standards (Pierce) and the bicinchoninic acid assay (Pierce). Total protein and phosphorylation levels of relevant proteins were assessed by standard Western blotting techniques. Primary antibodies used: AMPK Thr172 (Cell Signaling Technology (CST) #2531), ACC Ser79 (recognizes ACC2 Ser212 in murine skeletal muscle, CST #3661), ULK Ser555 (CST #5869), Akt Thr308 (CST #9275), Akt Ser473 (CST #9271), TBC1D4 Thr642 (CST #4288), ULK Ser757 (CST #6888). The membranes were blocked for 15 min at room temperature in TBS-Tween20 (TBST) containing either 3% BSA or 2% skimmed milk, followed by overnight incubation at 4 °C with the primary antibody. On the subsequent day, the blots wereincubated for 45 min at room temperature in the corresponding horseradish peroxidase-conjugated (HRP) secondary antibody and washed 3 times in TBST prior to imaging of the blots (ChemiDoc^TM^MP Imaging System, Bio Rad, Hercules, CA, USA).

### 4.6. TEM Analyses

A subset of EDL and soleus muscles (*n* = 3) were fixed with 2% glutaraldehyde in 0.1 M phosphate buffer (pH 7.4) for 5 h and stored in 0.1 M phosphate buffer containing 0.1% glutaraldehyde. Small superficial fiber bundles were teased free from the muscles and washed 3 times in 0.1 M sodium cacodylate buffer before incubation in 1% OsO_4_ for 1 h. The bundles were then stained with 3% uranyl acetate for 30 min. Dehydration was performed using graded concentrations of ethanol, and finally, the samples were transferred to propylene oxide and embedded in Epon. Ultra-thin sections were made using a Leica EM UC6 ultra microtome and these were counterstained with uranyl acetate and lead citrate. The sections were examined using a FEI 120 kV BioTwin T12 transmission electron microscope (FEI, Thermo Fisher Scientific, Waltham, MA, USA). Contrasting and cropping were performed in ImageJ 1.51u (National Institute of Health, USA).

### 4.7. GLUT4 Translocation Assay in L6 Muscle Cells

L6 myoblasts stably expressing GLUT4 with an exofacial c-myc epitope-tag were a kind gift from Dr. Amira Klip (SickKids Research Institute Toronto, University of Toronto, Canada). Myoblasts were seeded in a 96-well plate and kept in Minimum Essential Medium α (#22571020, Gibco) supplemented with 10% fetal bovine serum at 37 °C, and 5% CO_2_. On the experimental day, the confluent myoblasts were serum-starved for 4 h prior to incubation with or without 100 nM insulin for 30 min. For the last 30 min prior to insulin-stimulation and during insulin-stimulation, the cells were incubated with either DMSO or SBI-0206965 at the concentrations indicated in the figure. The cells were then briefly washed in ice-cold Dulbecco’s phosphate-buffered saline supplemented with Ca^2+^ and Mg^2+^ (DPBS) and fixed in 3% paraformaldehyde for 10 min on ice followed by 10 min at room temperature. The fixed cells were incubated for 45 min with Anti-c-myc antibody (C3956, Sigma), diluted 1:500 in 5% goat serum in DPBS, washed 3× in DPBS, and incubated with HRP-conjugated secondary anti-rabbit antibody (1:1000) for 30 min at room temperature. After washing again, O-phenylenediamine (OPD (P5412, Sigma)) reagent was added to start the colorimetric reaction. The reaction was stopped by adding 5 N hydrochloric acid and the optical absorbance was measured at 492 nm. The developed signal from a few wells without the primary antibody was subtracted as non-specific background [18].

### 4.8. Statistical Analyses

The results are presented as mean ± standard error of mean (SEM). The significance level was set to *p* < 0.05. Two-way analysis of variance (ANOVA) with Tukey’s post-hoc test was applied to identify differences between subgroups, if significant ANOVA effects were found. To inform the readers, we report *p*-values at or below 0.1 with the exact *p*-value presented. In data-sets, where the assumption of equal variance was violated, a natural log transformation was performed. The statistical analyses were carried out in SigmaPlot 14.

## Figures and Tables

**Figure 1 ijms-21-02344-f001:**
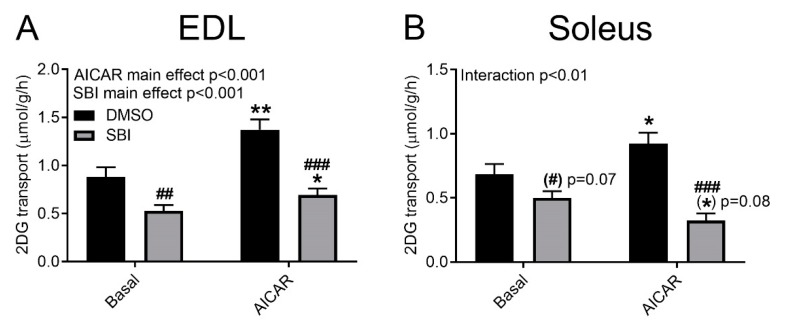
SBI-0206965 suppresses analog 5-aminoimidazole-4-carboxamide ribonucleotide (AICAR)-stimulated glucose transport. AICAR-stimulated (4 mM, 40 min) glucose transport with or without 10 µM SBI-0206965 for 1 h in (**A**) extensor digitorum longus (EDL) and (**B**) Soleus muscles. ANOVA main or interaction effects are indicated in the panels. */** *p* < 0.05/0.01 vs. Basal, ##/### *p* < 0.01/0.001 vs. DMSO. *n* = 20. All values are shown as mean ± SEM.

**Figure 2 ijms-21-02344-f002:**
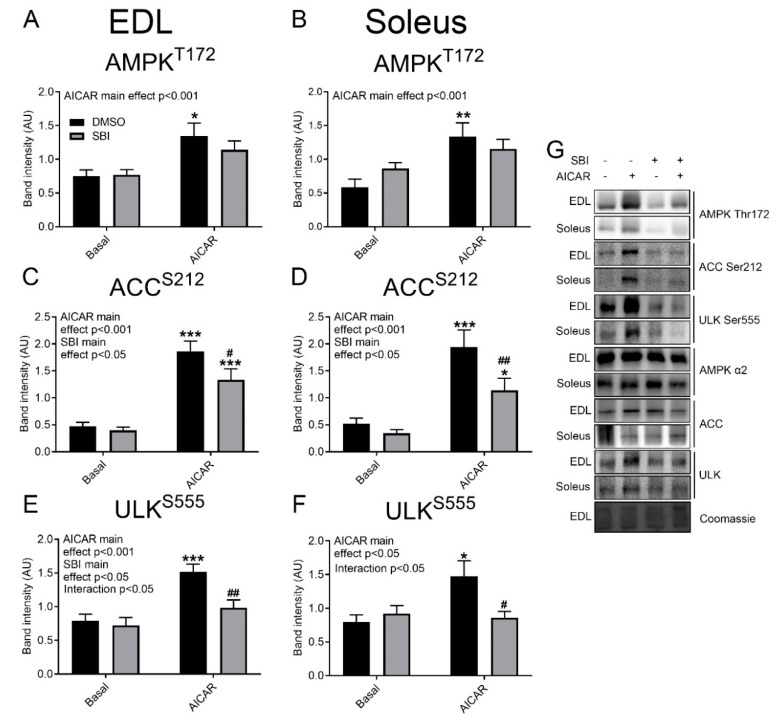
Modest impairment of AICAR-induced AMPK activation by SBI-0206965. Quantification of (**A**,**B**) AMPK Thr172, (**C**,**D**) ACC2 Ser212, and (**E**,**F**) ULK Ser555 phosphorylation in (**A**,**C**,**E**) EDL and (**B**,**D**,**F**) soleus muscles stimulated with or without AICAR (4 mM, 40 min) and with or without 10 µM SBI-0206965 for 1 h. (**G**) Representative blots of quantified and total proteins. ANOVA main or interaction effects are indicated in the panels. ^*/**/***^
*p* < 0.05/0.01/0.001 vs. Basal. ^#/##^
*p* < 0.05/0.01 vs. corresponding DMSO-treated group. A–E, *n* = 14–19, F, *n* = 7–8. All values are shown as mean ± SEM.

**Figure 3 ijms-21-02344-f003:**
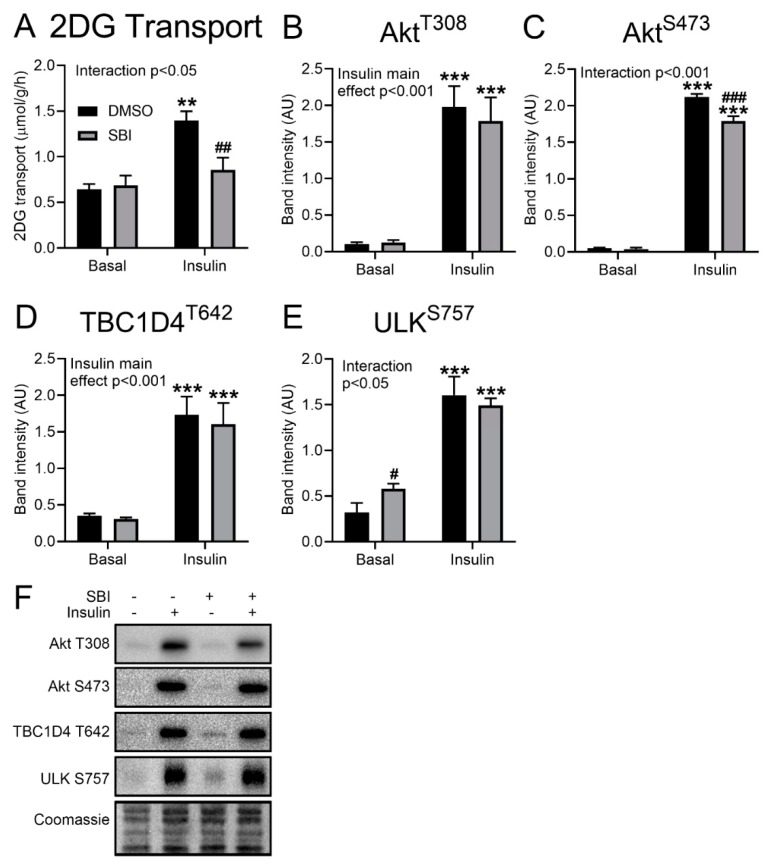
SBI-0206965 inhibits insulin-stimulated glucose transport. Quantification of (**A**) glucose transport and (**B**) Akt Thr308, (**C**) Akt Ser473, (**D**) TBC1D4 Thr642, and (**E**) ULK Ser757 phosphorylation in soleus muscles stimulated with or without insulin (60 nM, 20 min) and with or without 10 µM SBI-0206965 for 1 h. (**F**) Representative blots of quantified and total proteins. ANOVA main or interaction effects are indicated in the panels. ^**/***^
*p* < 0.01/0.001 vs. Basal, ^#/##/###^
*p* < 0.05/0.01/0.001 vs. DMSO. *n* = 3. All values are shown as mean ± SEM.

**Figure 4 ijms-21-02344-f004:**
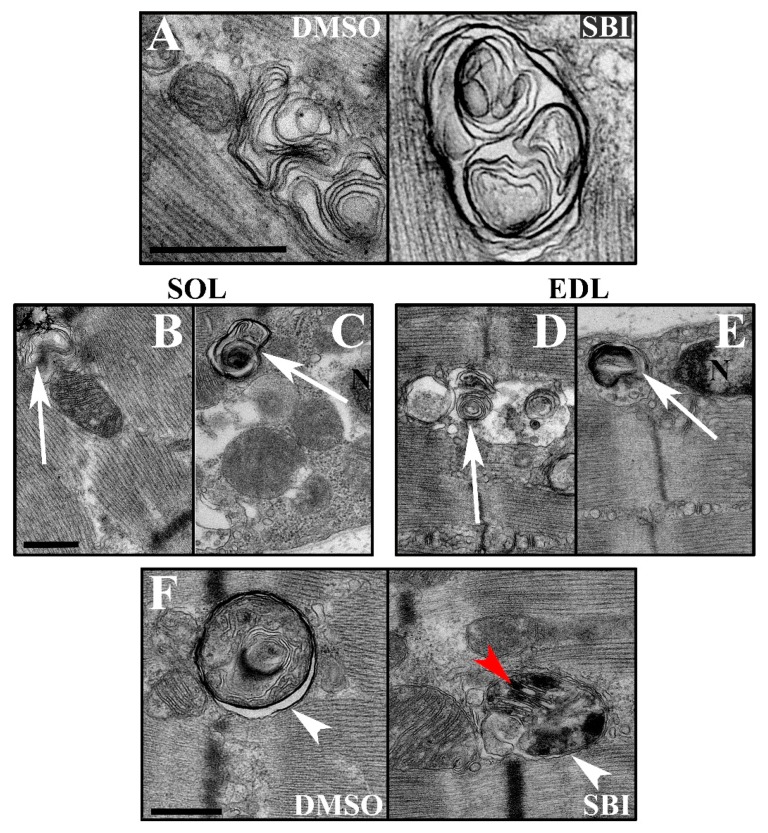
No evidence of abnormal membrane accumulation in SBI-0206965-treated muscles. (**A**) TEM micrographs displaying vacuoles with varying membrane layers from muscles incubated for 1 h with DMSO or 10 µM SBI. (**B**–**E**) The vacuoles (arrows) were observed in (**B**,**D**) intramyofibrillar and (**C**,**E**) perinuclear regions in both (**B**,**C**) soleus and (**D**,**E**) EDL muscles. (**F**) TEM micrographs displaying double membrane structures (white arrowheads) enclosing degraded material or organelles including mitochondria (red arrowhead) from muscles similar to A. N = Nucleus, scale bar = 500 nm. Representative images from muscle fibers from 3 different mice in each group.

**Figure 5 ijms-21-02344-f005:**
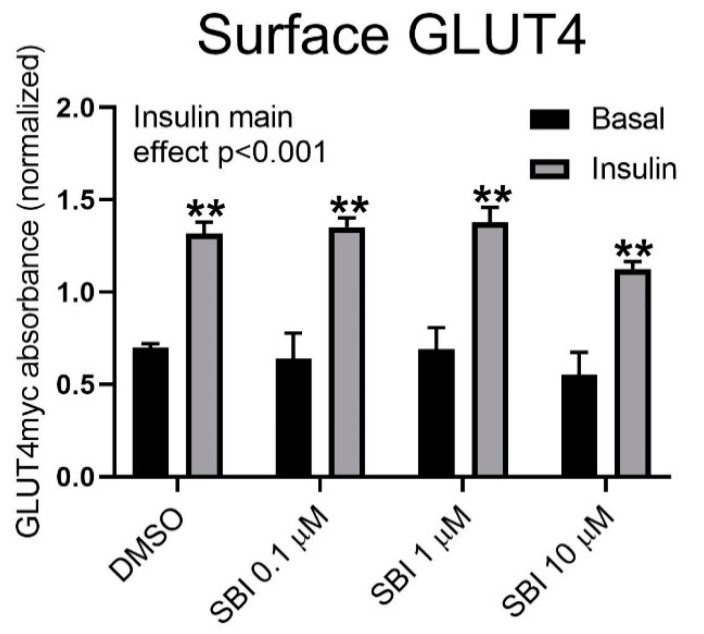
No effect of SBI-0206965 on GLUT4 translocation in L6 muscle cells. The quantified amount of GLUT4-myc in the surface membrane of L6 muscle cells treated with DMSO or SBI-0206965 and stimulated with or without insulin for 30 min. ANOVA main effect of insulin (*p* < 0.001). ** *p* < 0.01 vs. Basal. *n* = 2 from independent experiments. Each experiment is an average of 4 technical replicates. Values are shown as mean ± SEM.

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
