# Peer review of "The ULK1/2 and AMPK Inhibitor SBI-0206965 Blocks AICAR and Insulin-Stimulated Glucose Transport"

_ijms, 2020, doi:10.3390/ijms21072344_

Round 1

Reviewer 1 Report

The manuscript finds discrepancies in the specificity of a kinase inhibitor (abbreviated SBI). The results seem reasonable, and worthy of publication. However, there are a few points that need to be addressed.

  1. Statistics is not used properly. The point of having a cutoff prior to experimentation is to provide objectivity. Commonly that cutoff is p<0.05. Some use the dubious extension of showing when p is much less, demarking p<0.01, or less; that is followed here. It does not really add weight to a significance argument. But if that were the only problem, it could be overlooked. The more serious one is using values of p greater than 0.05 and marking them with various symbols to indicate "trends". This is improper. The citation of 0.06, 0.07, and 0.08, can be simply referred to as "not significant". It is essential that a cutoff be determined prior to data collection. 
  2. In one case, SBI stimulated basal ULK phosphorlyation (Fig.3E); this should be discussed in the discussion. If not, then the authors must consider that multiple nonselective actions of SBI are in force, and the compound is useless as an investigative tool, not that it be "used with caution" as the final paragraph in the discussion claims.
  3. The treatment of the muscle tissue in liquid nitrogen is troublesome, as it seems that the authors believe this quenches cellular reactions quickly. It does  not; liquid nitrogen forms bubbles when tissue is submerged and it takes several seconds for the tissue to freeze. It is not clear that this affects the results, but perhaps there was a precaution taken that was not mentioned in the text.
  4. 2-deoxyglucose transport method is not clear. In the cited reference,10, this method does not appear in the main text, and was not apparent at least to me in the supplemental document. A description of the method is in order since most readers will not find it either.

Reviewer 2 Report

In this manuscript, the authors investigated the specificity of SBI-0206965 in skeletal muscle, measuring the effect on AICAR-stimulated AMPK-dependent glucose transport and insulin-stimulated AMPK-independent glucose uptake. However, they should show the total AMPK and total ACC protein expression ã„´(in Fig 2,3 data) and GLUT4 in both membrane and cytosol. In addition, the authors showed that “SBI-0206965 partially supresses AMPK activation by AICAR” in Fig 2, thus you should be careful conclude “SBI blocks glucose transport independent AMPK activation”. I think this manuscript is not yet suitable for posting to IJMS.

Round 2

Reviewer 2 Report

They showed data that SBI-0206965 did not significantly inhibit AMPK activation in Fig 2, but indicated 'AMPK inhibitor SBI-0206965' in the manuscript. This revised manuscript is still difficult to conclude that AMPK and ULK1/2 inhibitor SBI-0206965 impairs muscle glucose transport. This manuscript is not suitable for publication.

Author Response

Reviewer 2: They showed data that SBI-0206965 did not significantly inhibit AMPK activation in Fig 2, but indicated 'AMPK inhibitor SBI-0206965' in the manuscript. This revised manuscript is still difficult to conclude that AMPK and ULK1/2 inhibitor SBI-0206965 impairs muscle glucose transport. This manuscript is not suitable for publication.

Reply
We have now increased the n in Fig. 2 and demonstrate a clear and significant reduction in AMPK activation by SBI-0206965.

We demonstrate that SBI-0206965 led to a complete blockade of muscle glucose transport induced by both AICAR (Fig. 1) and insulin (Fig. 3) in an experimental system excluding contributions from systemic factors. This is considered as strong evidence that SBI-0206965 impairs muscle glucose transport.

Round 3

Reviewer 2 Report

Now, this manuscript is suitable for publication in INTERNATIONAL JOURNAL OF MOLECULAR SCIENCES.